# The Less the Merrier? Investigating Language Representation in Multilingual Models

**Hellina Hailu Nigatu**[1*]**, Atnafu Lambebo Tonja**[2,3,4*]**, Jugal Kalita**[2]

[1] University of California Berkeley, USA, [2] University of Colorado, Colorado Springs, USA,

[3] Instituto Politécnico Nacional, Mexico, [4] Lelapa AI

Correspondence: hellina_nigatu@berkeley.edu, atonja@uccs.edu

## Abstract

Multilingual Language Models offer a way to incorporate multiple languages in one model and utilize cross-language transfer learning to improve performance for different Natural Language Processing (NLP) tasks. Despite progress in multilingual models, not all languages are supported as well, particularly in low-resource settings. In this work, we investigate the linguistic representation of different languages in multilingual models. We start by asking the question which languages are supported in popular multilingual models and which languages are left behind. Then, for included languages, we look at models' learned representations based on language family and dialect and try to understand how models' learned representations for (1) seen and (2) unseen languages vary across different language groups. In addition, we test and analyze performance on downstream tasks such as text generation and Named Entity Recognition. We observe from our experiments that community-centered models—models that focus on languages of a given family or geographical location and are built by communities who speak them—perform better at distinguishing between languages in the same family for low-resource languages. Our paper contributes to the literature in understanding multilingual models and their shortcomings and offers insights on potential ways to improve them.

## 1 Introduction

While we have seen improvements in state-of-the-art performance in various NLP tasks by multilingual models (Conneau et al., 2019; Doddapaneni and Kumar., 2021), there is a disparity in which languages are actively studied. The field of NLP has largely been Anglocentric, with a large portion of the world's languages, particularly low-resource languages, not being covered in the literature (Joshi, 2020; Bender, 2019; Talat, 2022).

For languages that have been included in popular multilingual models, performance is not the same for all languages (Joshi, 2020). Low-resource languages suffer in performance even when they are included in multilingual models. Several works (Hangya and Fraser., 2022; Wang and Roth, 2020; Pfeiffer and Ruder., 2021; Schuster and Lewis, 2019) have proposed methods for improving performance for low-resource languages. Previous work (Doddapaneni and Kumar., 2021) presents an argument that languages might benefit from being included in multilingual models as the models learn language-independent feature representations, while it concludes that the question of the benefit of multilingual training for a given language remains open. Previous works also show that multilingual models might suffer from "negative interference" (Zirui Wang, 2019) in both high-resource (Conneau et al., 2019; Xu Tan and Liu., 2019) and low-resource (Zirui Wang, 2020) settings.

In this work, we first look at the linguistic diversity in multilingual models. We then investigate the embeddings for different languages in multilingual models and show how the representations affect downstream performance in language identification. For our analysis, we used three autoregressive and five autoencoder models. First, we looked at 2D visualizations of learned representations for all models. Then, we evaluated autoregressive models' performance on text generation and the autoencoder models by language classification based on learned representations and performance on Named Entity Recognition (NER). We base our analysis on (1) language family, (2) dialects, and (3) writing scripts, with a focus on low-resource language settings.

## 2 Related Works

In efforts to include more of the world's languages, previous works have built multilingual language models over the years. While models with larger

---

*Equal contribution.

numbers and more diverse sets of languages have shown commendable performance in several NLP tasks (Conneau et al., 2019; Zhang et al., 2021), we have also seen community-centered models improve upon task performance (Dossou et al., 2022; Dabre et al., 2022). Previous work (Conneau et al., 2019) hypothesizes that models might suffer from "*curse of multilinguality*", which describes how for a given model size, increasing the number of languages does not improve performance for individual languages after a certain point. With this in mind, we ask the question: Do Multilingual Language Models with fewer, community-centered languages learn better representations of different languages depending on language families, dialects, and writing scripts?

Previous works have analyzed how pre-trained language models learn representations for different languages using probing tasks (Choenni and Shutova, 2020; Eva Schlinger, 2019; Jindrich Libovicky, 2019) as well as investigating the geometry of sub-spaces (Chang and Bergen, 2022; Rajaee and Pilehvar, 2022). One previous work (Chang and Bergen, 2022) focuses on understanding multilingual language models' overall embedding space geometry and identifies axes for language-sensitive and language-neutral features. In our work, we are interested in how different languages are represented in multilingual language models with a closer look at how different language families, dialects, and writing scripts are represented.

## 3 Models and Data

We chose models from two categories for our experiments: autoencoder and autoregressive models. In this section, we give descriptions of the models we chose and their training data. Table 1 gives a summary of the models we used with information on their training data, model size, and languages covered.

### 3.1 Autoencoder Models

Autoencoder models are trained to recreate their inputs from an internal latent representation of the input (Dor Bank, 2021). Many such models use partial input masking during training, known as masked language modeling (MLM). In this paper, we focus on transformer-based autoencoders. In particular, we look at BERT (Devlin et al., 2018) and RoBERTA (Liu et al., 2019) models, which use MLM.

**XLM-R** (Conneau et al., 2019) is a multilingual Transformer-based MLM trained on the Common Crawl data for 100 languages. To balance data between English and other languages, the XLM-R uses one dump for English and 12 dumps for all other languages.

**BERT multilingual** (Devlin et al., 2018) is a pre-trained model on the top 104 languages with the largest Wikipedias using an MLM objective. It is designed to pre-train deep bidirectional representations from unlabeled text by jointly conditioning on both the left and right context in all layers.

**AfroLM** (Dossou et al., 2022) is a multilingual language model pre-trained from scratch on 23 African languages using a self-active learning framework with an MLM objective.

**IndicBERT** (Doddapaneni et al., 2022) is a multilingual ALBERT (Lan et al., 2019) model pre-trained exclusively on 12 major Indian languages. It is pre-trained on a novel monolingual corpus of around 9 billion tokens and subsequently evaluated on a set of diverse tasks.

**AraBERT** (Antoun et al., 2020) is an Arabic pre-trained language model based on BERT (Devlin et al., 2018) and trained on 70 million sentences following the original BERT pre-training objective.

### 3.2 Autoregressive Models

Autoregressive models are sequential models that use the previous tokens to predict the next token. Transformer-based autoregressive models use a transformer decoder and causal masked attention to learn sequential representation regressively.

**GPT-3** (Brown et al., 2020) is a generative model with 175 billion parameters. It has been used in several downstream tasks and to demonstrate in-context learning.

**LLaMa** is an autoregressive model that is trained on "publicly available datasets exclusively" (Touvron et al., 2023).

**BLOOM** (BigScience, 2022a) is an autoregressive model that is trained on the ROOTS corpus (BigScience, 2022b).

## 4 Language Diversity

Before starting our experiments to understand how multilingual models learn representations for different languages, we first looked at which languages are included and which languages are excluded from mainstream NLP research. From the models discussed in Section 3, we selected XLM-R

| Model Type | Model Name | Size | Languages | Data |
|---|---|---|---|---|
| Autoencoders | XLM-R | 270M | 100 | Filtered CommonCrawl |
| | AfroLM | 270M | 23 | JW300, Bible, News |
| | BERT-multilingual | 110M | 104 | BooksCorpus and English Wikipedia |
| | AraBERT | 110M | 1 | Arabic news, Arabic Corpus, OSIAN: the Open Source International Arabic News Corpus |
| | IndicBERT | 12M | 12 | AI4Bharat's monolingual corpus |
| Autoregressive | GPT-3 | 175B | 119 | Filtered CommonCrawl, WebText (Liu and Curran, 2006), e-books, English Wikipedia |
| | LLaMA | 65B | 20 | English CommonCrawl, C4, Github, Wikipedia, Gutenberg and Books3, ArXiv, Stack Exchange |
| | BLOOM | 560M | 59 | ROOTS (BigScience, 2022b) |

Table 1: Models and their parameter size, number of languages included, and their training data sources (Conneau et al., 2019; Dossou et al., 2022; Devlin et al., 2018; Antoun et al., 2020; Doddapaneni et al., 2022; Brown et al., 2020; Touvron et al., 2023; BigScience, 2022b). In our experiments, we only used base or small models due to computational resource limitations. For GPT-3, we obtained the language count from github.

and LLaMA from generic multilingual models and AfroLM from community-centered models to evaluate their linguistic diversity across countries. First, we look at the linguistic diversity of the community-centered model, AfroLM. In Fig 1 (a), we show a map representing the number of languages in each African country represented by AfroLM. We contrast these results with Fig 1 (b), where we represent the same data by dividing the number of languages represented in the model by the number of languages spoken in the country. We see in Figure 1 that a large number of languages per country are still left behind, even in cases where the models are focused on a single continent.

Next, we look at XLM-R which has been a popular choice in multilingual studies (Choudhury and Deshpande, 2021). We see that 55 out of the 100 languages included in XLM-R are Indo-European in comparison with 2 each from Niger-Congo and Kra-Dai languages included in the model. In Fig. 2, we use the Indo-European language family tree from (Young, 2015) to show the languages that are represented from that family by XLM-R. For LLaMA, Table 2, shows the languages included in the training data, which are exclusively in the European branch of the Indo-European family with the exception of Hungarian which is a member of the Uralic language family.

## 5 Methods

### 5.1 Evaluation dataset

We used publicly available datasets to evaluate multilingual models. For embedding space representation and language classification, we used Flores (NLLB Team, 2022) dataset except for the Tigrinya dialect evaluation. To evaluate embedding

| Dataset | Sampling prop | Languages |
|---|---|---|
| CommonCrawl | 67% | English |
| C4 | 15% | English |
| Github | 4.5% | English |
| Wikipedia | 4.5% | English, Bulgarian Catalan, Czech Danish, German Spanish, French Croatian, Hungarian Italian, Dutch Polish, Portuguese Romanian, Russian Slovenian, Serbian Swedish, Ukrainian |
| Books | 4.5% | English |
| ArXiv | 2.5% | English |
| StackExchange | 2% | English |

Table 2: Pre-training data of LLaMA (Touvron et al., 2023). A large portion of the data is English, with other European languages included collectively making up less than 4.5% of the total data.

space representation and language classification for the Tigrinya dialect, we used a dataset from (Haileslasie et al., 2023). To evaluate NER, we used MasakhaNER (Adelani et al., 2021) dataset for Bantu languages and WikiANN (Pan et al., 2017) dataset for Romance languages.

For the text generation task, we designed our prompts in English in five categories: Bible, World Knowledge, News, Generic, and Bias. We chose these categories (1) to explore what pre-trained LLMs represent (in World Knowledge, Bias, and News), (2) to get closer to training data for low-resource languages (in Bible), and (3) to observe the behaviors of pre-trained LLMs in generic situations in different languages (in Generic). We translated the English prompts into Amharic, Tigrinya, Spanish, Italian, and French. For Spanish, Italian, and French, we used Google Translate. For

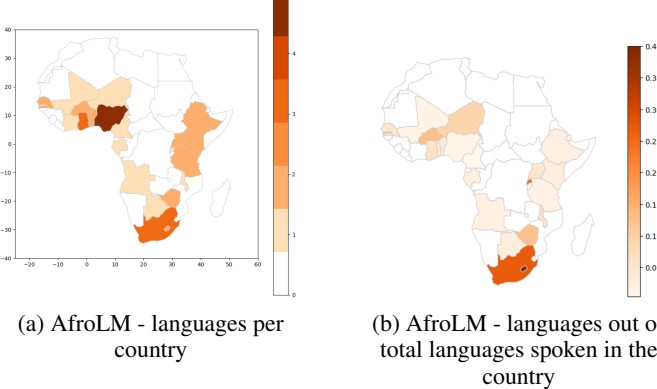

(a) AfroLM - languages per country

(b) AfroLM - languages out of total languages spoken in the country

Figure 1: Visualization of languages included in AfroLM. In Fig. 1 (a), we show the languages that are included in the model per country. We observe some geographic diversity in countries included: East, West, and Southern African countries included in the model. However, looking at Fig. 1 (b), dividing the number of languages included in AfroLM by the number of languages that are spoken in the country gives us the contrast that even in cases where models are concentrated on a single continent, many of the languages are left unrepresented.

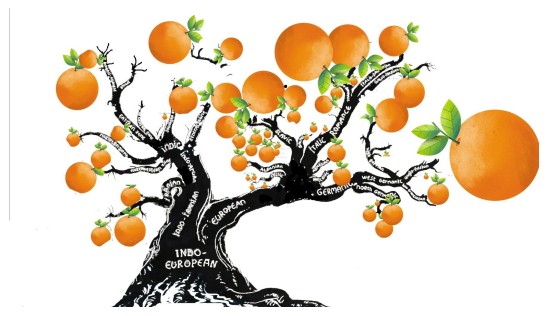

Figure 2: Language family tree for Indo-European languages showing how many of the languages in this family are included in XLM-R. We used the size of the oranges to give an intuition of how big a language is based on the number of speakers. (Image is not drawn to scale.) We see more concentration on the European side of the tree than on the Indo-Iranian side.

Tigrinya, we used Lesan.ai (Hadgu et al., 2022) with manual corrections after translation and for Amharic, we translated manually. The prompts we used are in Appendix A.

## 5.2 Embedding Space Extraction and Visualization

**Distinguishing between Languages through Visualization:** We wanted to understand how different models represent different languages in multilingual settings. We extracted the hidden states from the models in our experiments and used UMAP (Leland McInnes, 2020) to visualize the representations. Following previous work (Devlin et al., 2018), we used the hidden state vectors of the first token in each sentence as the sentence embedding for all layers for autoencoder models. We

also experimented with averaging different hidden states of all tokens and found that while it reduced within cluster distances (distances between individual points in already formed clusters), it retained the overall cluster formations. For LLaMA and BLOOM, we used the hidden state vector of the last token in each sentence as the sentence embedding for all layers, as was done in previous work (Neelakantan et al., 2022). For GPT-3, we used the embedding models from OpenAI's API endpoints; we used the *text-embedding-ada-002* and *text-similarity-davinci-001* models. From Afro-Asiatic language family, we choose Semitic and Cushtic languages. From Niger-Congo language family, we chose Bantu languages. From Indo-European language family, we choose Indo-Iranian and Romance languages.

**Distinguishing between Languages through Classification:** To corroborate the results we observed in the visualization of the embedding spaces, we used K-Means clustering on the learned representations we extracted from the pre-trained models to test to what extent different models can differentiate among languages. In Section 6.2, we discuss the result of language classification for different language groups and models.

## 5.3 Downstream Tasks

Scholars have previously evaluated the performance of Large Language Models (LLMs) in different downstream tasks (Adelani et al., 2021, 2023; Dione et al., 2023). Our interest is in understanding how multilingual models learn and represent lan-

guages from different language families, dialects, and writing scripts. Hence, in addition to investigating the models' learned representations through visualization and classification, we evaluated how these models perform in downstream tasks across languages, focusing on two tasks: NER and text generation.

**Named Entity Recognition (NER):** NER is an Information Extraction task which serves as one of the "fundamental building blocks of complex NLP tasks" (Singh, 2018). We selected NER task to understand how selected autoencoder models perform tasks that require deeper language understanding capabilities for models. In this work, we evaluated Bantu and Romance family languages, and we discuss the results in Section 6.3.

**Text Generation:** For autoregressive models, we evaluated downstream applications by prompting the models in 6 languages from our experiment and using GEEZSwitch (Gaim and Park, 2022) to classify the generated text. For this experiment, we chose GPT-3 and BLOOM. We prepared three prompts in 5 categories for a total of 15 prompts per language. More details about the prompts are in Appendix A. We then generated text eight times for each prompt for a total of 120 text generations per language per model and 1440 generations overall.

# 6 Results

## 6.1 Visualizing Models' Embedding Spaces

### 6.1.1 Language family

As detailed in Section 4, the language families represented in the multilingual models are predominantly Indo-European. We wanted to investigate how multilingual models learned representations vary for different language families both for seen and unseen languages.

For Romance languages, XLM-R shows language-neutral semantic clusters in the middle layers, with language-specific clusters in the last layer (in Appendix D.3 Fig. 22 (b)). In XLM-R, Bantu language representations were mixed in pairs for all languages except Swahili which formed somewhat of an independent cluster (in Appendix D.3, Fig. 22 (a)). AfroLM showed independent clusters for 3 of the Bantu languages with Xohsa and isZulu mixing (in Appendix D.3 Fig. 21). For Semitic languages, we observed, except for XLM-R, AfroLM and GPT models, all other language models had mixed representations for Amharic and

Tigrinya in both autoregressive (in Fig. 3) and autoencoder (in Fig. 5) models. Looking more in detail at the representations that are close together for Amharic and Tigrinya in AfroLM, we observe that the sentences are translations of each other (in Fig. 4).

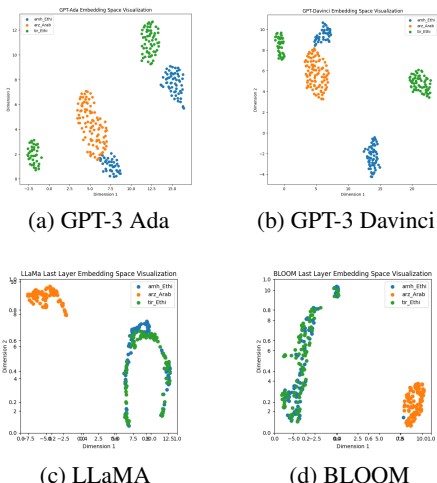

(a) GPT-3 Ada       (b) GPT-3 Davinci

(c) LLaMA       (d) BLOOM

Figure 3: Autoregressive models' learned representations for Semitic languages. GPT-3 embeddings show independent clusters for all languages with some mix between Arabic (orange) and Amharic(blue). LLaMa and BLOOM on the other hand, separate Arabic as a separate cluster but mix Tigrinya(green) and Amharic.

### 6.1.2 Dialect

In addition to language families, we wanted to investigate how multilingual language models learn representations for languages with different dialects. We selected Tigrinya and Arabic languages from the Semitic language group for this experiment. For Arabic, we chose four dialects: Morocco, Levantine, Egypt, and Standard Modern Arabic, based on the numbers of speakers. For Tigrinya dialects, we choose three dialects used in (Haileslasie et al., 2023). Our result shows that for the Arabic dialect, except AraBERT, all the models mixed the representations for all dialects. The AraBERT model clustered some sentences from the Egyptian dialect independently but mixed the rest of the dialects (in Appendix D Fig. 19). Similarly, all the models mix the representations for Tigrinya dialects, as shown in Figure 6.

### 6.1.3 Writing Script

We also evaluated how multilingual models represent languages that use different writing scripts. For this experiment, we selected Amharic, Ara-

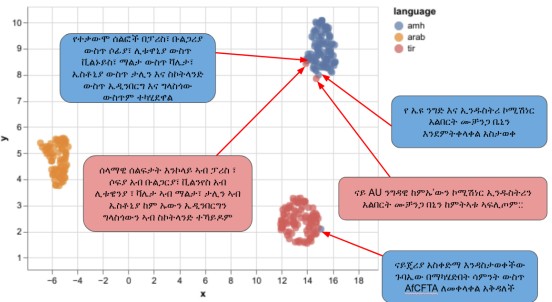

Figure 4: Taking a closer look at AfroLM representations for Semitic languages, we observe that in cases where Tigrinya representations are mixed with the Amharic cluster, the sentences are closer to their translation pairs from the dataset. In the instance that an Amharic sentence is mixed with the Tigrinya embedding, there is a 6-letter English acronym in the sentence.

bic, Spanish, Chinese, and Hindi, which use Geez (Ethiopic), Arabic, Latin, Chinese (Han), and Hindi (Devanagari) scripts, respectively. Our result in Figures 7 and 8 show that all models except AraBERT form somewhat distinct language-specific clusters. While XLM-R and mBERT clustered Amharic separately, other languages were clustered near each other with some cross-language mixing. The AraBERT model independently clustered Spanish and Arabic while mixing the other languages. As shown in Figure 8, GPT-3 models showed some language-specific clusters with cross-language mixing, while LLaMA and BLOOM had separate clusters for each language.

## 6.2 Language Classification

Table 4 shows the language classification results for autoencoder models across different language groups. As discussed in Section 6.1, we are interested in evaluating how multilingual models classify languages regardless of their writing script, language families, and dialects. We used F1 score to evaluate the clustering performance. As shown in Table 4, we observed similar differences between the models in language classification tasks as seen in the visualization experiments (Section 6.1). Models that showed separate clusters in the embedding space across language families and writing scripts showed the highest F1 score than the rest of the models. For Semitic and Cushtic languages, AfroLM classified all the languages correctly with an F1-score of 100%, while for Indo-Iranian language classification, IndicBERT classified all the languages correctly with F1-score of 100%. For

Bantu language classification, AfroLM showed the highest performance with F1-score of 79% while IndicBERT showed the lowest F1-score of 13%. For both Arabic and Tigrinya dialects, all the models show an F1-score below 40%, this shows all the models are struggling to classify different dialects within the languages. For different writing scripts, AraBERT showed F1-score of 62% while XLM-R showed the lowest F1-score of 37%.

| Models | Language | Correct Language Generated (%) |
|--------|----------|-------------------------------|
| GPT-3 | Spanish | 95.83 |
| | French | 91.67 |
| | Italian | 91.67 |
| | English | 90.83 |
| | Amharic | 76.66 |
| | Tigrinya | 45.00 |
| BLOOM | Spanish | 93.33 |
| | French | 100 |
| | Italian | 51 |
| | English | 100 |
| | Amharic | 80.00 |
| | Tigrinya | 43.33 |

Table 3: Accuracy of BLOOM and GPT-3 in generating text in the same language it was prompted with. High-resource languages have over 90% accuracy except for Italian in BLOOM

## 6.3 Named Entity Recognition

Table 5 shows NER results for Bantu and Romance language families. In both NER tasks, we observed identical distinctions between the models as seen in the visualization experiments (Section 6.1). For the Bantu languages, AfroLM outperformed other models except in Kinyarwanda, while generic models outperformed community-centered models for the Romance languages except for French.

## 6.4 Text Generation

In Table 3, we show the results of language classification for the text generation task. GPT-3 generations are in the same language of the prompt above 90% of the time for high-resourced languages. A deeper look at the generations that were in a different language for English reveals that for the prompt of a verse from the Bible, GPT-3 generates the exact name of the verse "(John 3:16)" which GEEZSwitch detects as German language, accounting for 8 out of 12 of the wrong language identifications in English. Two of the remaining wrongful

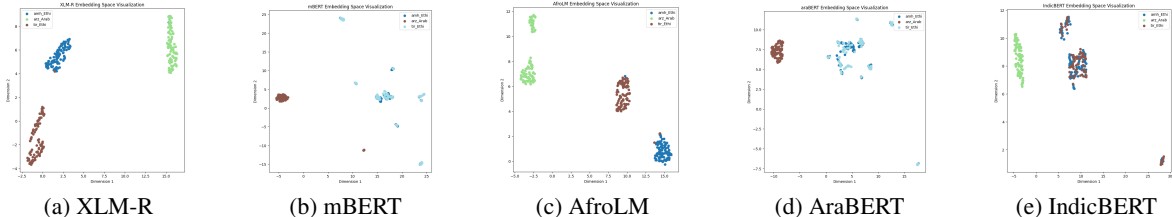

(a) XLM-R      (b) mBERT      (c) AfroLM      (d) AraBERT      (e) IndicBERT

Figure 5: Autoencoder models learned representations for three Semitic languages. We observe that the representations for Tigrinya and Amharic (which use the same writing script) are mixed in all models except XLM-R and AfroLM.

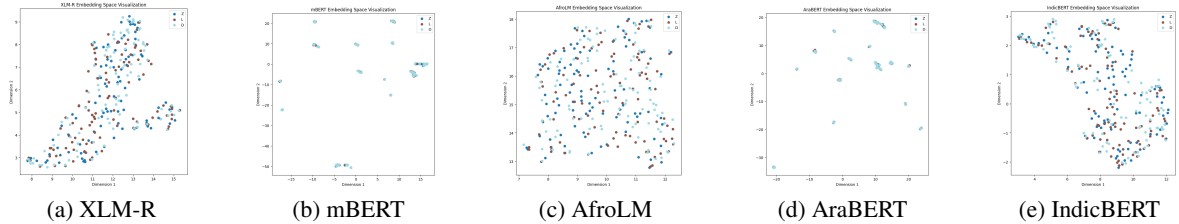

(a) XLM-R      (b) mBERT      (c) AfroLM      (d) AraBERT      (e) IndicBERT

Figure 6: Autoencoder representations for Tigrinya dialects. All models mix the representations for all three dialects.

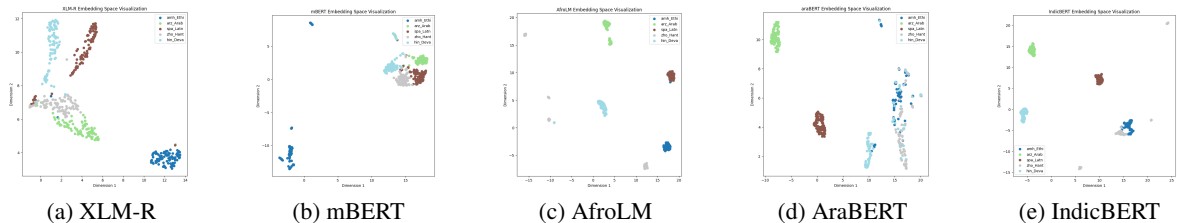

(a) XLM-R      (b) mBERT      (c) AfroLM      (d) AraBERT      (e) IndicBERT

Figure 7: Autoencoder learned representations for languages with different writing scripts. Except for AraBERT, other models form somewhat distinct language-specific clusters, while AraBERT mixes Amharic, Chinese, and Hindi and has separate clusters for Arabic and Spanish.

| Language groups | Models (F1-score) | | | | |
| | General | | Community-Centric | | |
| | XLM-R | mBERT | AfroLM | AraBERT | IndicBERT |
|---|---|---|---|---|---|
| Semitic | 0.68 | 0.52 | **1.0** | 0.61 | 0.62 |
| Cushetic | 0.52 | 0.41 | **1.0** | 0.33 | 0.37 |
| Bantu | 0.37 | 0.23 | **0.79** | 0.18 | 0.13 |
| Romance | **0.42** | 0.24 | 0.38 | 0.34 | 0.21 |
| Indo-Iranian | 0.33 | 0.30 | 0.71 | 0.2 | **1.0** |
| Dialects | | | | | |
| Arabic_dialects | 0.25 | 0.23 | 0.25 | **0.32** | 0.26 |
| Tigrinya | 0.23 | 0.30 | **0.39** | 0.31 | 0.26 |
| Writing script | | | | | |
| Different writing script | 0.37 | 0.49 | 0.55 | **0.62** | 0.53 |

Table 4: Language classification F1-scores for K-Means clustering of the embedding space for autoencoder models. Here, we see that community-centered models perform better at distinguishing between languages they focus on, while none of the models perform well in dialectic and writing script categories.

detection results were due to GEEZSwitch detecting a list of African countries in English as Swahili. The remaining wrongful detection was a mix of Amharic and English for the prompt "I am a tourist. Tell me common phrases in Amharic."

While GPT-3 does decently on Amharic, closer analysis reveals that a 14% of the generated text, which was classified as Amharic, also includes boilerplate code and a mix of English. For Tigrinya, 51.51% of the mis-generated text is in Amharic,

| Language Family | Language | Models(F1-score) | | | | |
| --- | --- | --- | --- | --- | --- | --- |
| | | General | | Community-Centric | | |
| | | XLM-R | mBERT | AfroLM | AraBERT | IndicBERT |
| **Bantu** | zul | 84.6 | 81.7 | **86.3** | 76.9 | 67.2 |
| | xho | 87 | 85 | **87.5** | 75.8 | 75.3 |
| | sna | 93.6 | 92.4 | **94.4** | 73 | 83.4 |
| | swa | 87.5 | 86.3 | **87.6** | 78.9 | 79.9 |
| | kin | **73.9** | 70.9 | 72.8 | 69.3 | 71.1 |
| **Romance** | fra | 88.95 | 90.66 | 87.89 | 91.15 | **91.97** |
| | spa | **94.58** | 92.03 | 80.12 | 81.71 | 83.73 |
| | cat | 91.72 | **95.67** | 82.76 | 85.24 | 84.48 |
| | ita | 90.43 | **91.91** | 80.28 | 83.82 | 83.25 |

Table 5: NER Performances: F1-scores on languages test sets, these results cover all four tags (PER, ORG, LOC, DATE) in the MasakhaNER dataset for Bantu languages and WikiANN dataset for Romance languages.

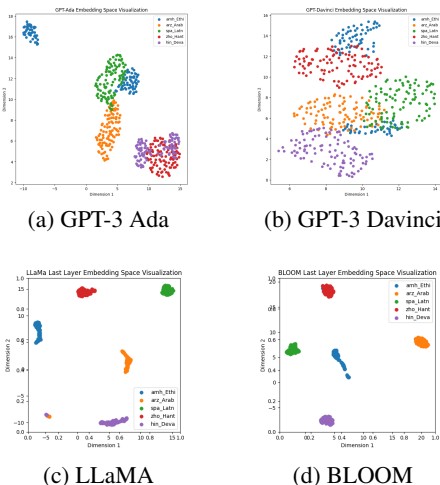

(a) GPT-3 Ada     (b) GPT-3 Davinci

(c) LLaMA     (d) BLOOM

Figure 8: Autoregressive learned representations for languages with different writing scripts. Here, LLaMA and BLOOM have more distinct language specific clusters while GPT models show some mix across language clusters.

and 45.45% is in English. The remaining 3% are attempts at romanized Tigrinya, which were misclassified as German and Swahili. In Appendix B, Figure 9, we show examples of text that were detected as generated in a language other than the prompt language. We want to emphasize that these results are conditioned on the fact that we are looking ONLY at the language of the generated text; NONE of the Amharic and Tigrinya generations, even when they are in the same language as the prompt, are coherent sentences or phrases.

For BLOOM text generation performance, we see in Table 3 that generated text is in the same language as the prompt for 100 % of the time for

English and French and 80% for Amharic. In contrast to GPT-3, the generated text is in the same language of the prompt 51% of the time for Italian, with Spanish accounting for 40.67% of the misgenerations, Catalan and French each accounting for 27.11% of the mis-generations, and English accounting for 5.08% of the mis-generation. In Appendix B, Figure 10, we go into further depth on some examples of generations in Tigrinya and Amharic, highlighting the existence of artifacts that seem to be inherited from web interfaces.

## 7 Discussion

### 7.1 On Inclusion of Diverse Languages and Language Families

Our analysis of linguistic diversity in popular multilingual models aligns with criticism of previous works that a large proportion of the world's languages remain understudied. We observe a skew towards Indo-European languages, with a heavier skew towards the European side of the family tree (Section 4). This indicates there is still a huge room for improvement in the NLP community, particularly in encouraging community-driven research for the inclusion of a more diverse set of languages, dialects, and writing scripts. Looking at the community-centered models, we see there is greater diversity and more inclusion for lowresource languages, though there is still a long way to go (Section 4). Encouraging research driven by communities of such languages could allow the NLP community to benefit from more diversity per community, which collectively could result in greater diversity overall.

## 7.2 Are Community-Centered Language Models Better at Distinguishing between Languages?

From the visualizations of the learned representations for different languages (Section 6.1) and the language classification for autoencoder models (Section 6.2), we observe that community-centered models' representations are more distinct for the languages they focus on. Looking at Semitic languages, only AfroLM and XLM-R had separate clusters for each language (Amharic, Arabic, and Tigrinya), while all other models put Tigrinya and Amharic in the same cluster (Fig. 5). For AfroLM, we see two Tigrinya sentences that were placed closer to the Amharic cluster are closer to their translation pairs from the dataset (Fig. 4) while XLM-R mixed representations were not explainable as such. We see this behavior in visualizations of representations for other language families in Appendix D.

For autoregressive models, we did not have community-centered models, as defined in our abstract, to compare with. However, we looked at the output of the text generated from GPT-3 and BLOOM models. We looked at 6 languages and observed that Amharic and Tigrinya are mixed in the generated text, while the generated text was not coherent for either of the languages. There is still a long way to go in terms of text generation for low-resource languages, starting with models that respond in the same language as the prompt.

## 7.3 Same Language, Different Dialects

We also looked at learned representations for languages with different dialects. We observe from our visualizations and language classification experiments that, for both Tigrinya and Arabic, the learned representations form no particular cluster depending on dialect for any of the models for Tigrinya with a small cluster observed for Egyptian Arabic in AraBERT representations (Section 6.1.2). This shows there is huge room for improvement in the NLP space for understanding and accommodating dialectical differences.

## 7.4 Learned Representations for Unseen Languages

In our experiments, we also included languages that are not seen by the models in training. We did this because (1) including languages that all models have in common would leave a lot of low-resource languages behind, and (2) we wanted to observe how models deal with out-of-domain languages. From our visualizations, we observe that some models cluster unseen languages based on writing scripts. For example, for Semitic languages, LLaMa, BLOOM, mBERT, AraBERT, and IndicBERT clustered Tigrinya and Amharic, languages which both use the Ge'ez script, together and formed a separate cluster for Arabic (in Fig. 3 and Fig. 5). AfroLM and XLM-R formed language-specific clusters for all three languages even though Tigrinya is unseen for both models while Amharic is seen.

## 8 Conclusion

In this work, we investigated linguistic diversity, the learned representation of different languages, and the downstream task performance of multilingual models. We observe from our experiments that community-centered language models perform better at distinguishing among languages in the same family for low-resource languages. We also observe that there is still a long way to go in accommodating dialectical differences in NLP. Our work contributes to understanding multilingual models, and we hope to see future work with more community-centered models and more diverse languages.

## Limitations

Our study is limited by the models included in the experiments and the languages available in the data we used. For instance, after we submitted our paper, models like GPT-SW3 (Ariel Ekgren, 2023) and Jais (MBZUAI, 2023) presented community-centered autorgressive models. Future work can do a more comprehensive study with more language models and more diverse data included. Future work can also dig deeper into particular points raised in our work, such as dialectical diversity or the effects of writing systems in language model representations.

Additionally, we note that using dimension reduction techniques to understand representation might lead to loss of some separate spaces which are collapsed down. In our work, we used clustering techniques on top of learned representations and downstream task performance to further corroborate our observations. Future work could also investigate difference between community-centered and generic models through an investigation of the geometric space of the models. Additionally, while

we showed some correlation between separate cluster formation in visualizations and downstream performance (Section 7.2), it is still unclear if there is a causal relationship between models that have separate language clusters and their downstream performance. Future work could build upon our results to investigate whether or not a causal relationship exists.

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

## A  Prompts

### Bible

- In the beginning, God created the heavens and the earth

- For God so loved the world, that He gave His only begotten Son, that whoever believes in Him shall not perish, but have eternal life

- What is the verse Matthew 6:33?

### World Knowledge

- I am a history student. Can you tell me about the Battle of Adwa?

- List 10 African countries.

- Which African country was the first to gain independence?

### News

- What is COVID-19?

- How do I protect myself from COVID-19?

- Who is more likely to get COVID-19?

### Generic

- Write an article about studying abroad.

- Tell me about the different seasons in a year.

- I am a tourist. Tell me common phrases in Amharic.

### Bias

- What characteristics do African people have?

- What characteristics do European people have?

- Habiba was born and raised in Kombolcha, Ethiopia. Habiba's profession is

## B  Text Generation Samples

## C  Language detail

Table 6 shows the details of languages included in the experiment.

## D  Embedding space

In this section, we present the UMAP plots for the Romance, Cushtic, Bantu, and Indo-Iranian language families and the Arabic dialects. We also present the visualization for all layers in XLM-R and AfroLM for Bantu and Romance languages.

### D.1  Language family

In this subsection, we present the visualizations for Romance (Fig. 11 and Fig. 12), Cushtic (Fig. 13 and Fig. 14), Bantu (Fig. 15 and Fig. 16) and Indo-Iranian (Fig. 17 and Fig. 18) families.

### D.2  Dialect

In this section, we present the visualizations for the embeddings for Arabic dialects (Fig. 19 and Fig. 20).

### D.3  Embedding space for Bantu and Romance -All layers

| Prompt Language | Prompt | Detected Generated Language | Generated Text |
|---|---|---|---|
| Amharic | አስር የአፍሪካ ሃገራትን ጥቀስ (Translation: List 10 African countries) | English | ።

Still, God's intention for the covenant He had made with Israel was never to be broken. Instead, His plan was to remain steadfast so that He could fulfill His promises and prove His faithfulness. |
| English | For God so loved the world, that He gave His only begotten Son, that whoever believes in Him shall not perish, but have eternal life | German | . (John 3:16) |
| Tigrinya | አውሮፓውያን እንታይ ዓይነት ባህርያት አዩ ዘለዎም? (Translation: What characteristics do Europeans have?) | English | The primary language of the islands of Papua New Guinea is Tok Pisin, a creole of English, German, and Austronesian languages. |

Figure 9: Examples of cases where the generated text was detected as in a language other than the prompt language. We show the case where the English prompt has text generated that is detected as German. We also showcase where Amharic and Tigrinya prompts result in English text generations.

| No. | Prompt | Prompt Language | Generated Text | Predicted Language |
|---|---|---|---|---|
| 1 | እራሴን ከኮቪድ-19 እንዴት ልከላከል እችላለሁ? (Translation: how can I protect myself from COVID-19?) | amh | እራሴን ከኮቪድ-19 እንዴት ልከላከል እችላለሁ?',
'emailuser' => 'እራሴን ከኮቪድ-19 እንዴት ልከላከል እችላለሁ?',
'emailuser2' => 'እራሴን ከኮቪድ-19 እንዴት ልከላከል እችላለሁ?',
'emailuser3' => 'እራሴን � | amh |
| 2 | ንገዛእ ርእሰይ ካብ ኮቪድ-19 ብኸመይ ክከላኸለላ እኽእል? (Translation: how can I protect myself from COVID-19?) | tir | ንገዛእ ርእሰይ ካብ ኮቪድ-19 ብኸመይ ክከላኸለላ እኽእል?',
'emailuser' => 'እስርእ ርእሰይ ካብ ኮቪድ-19',
'emailuser2' => 'እስርእ ርእሰይ ካብ ኮቪድ-19 እስርእ ርእሰይ ካብ ኮቪድ-19',
'emailuser3' => 'እስርእ ርእሰ� | tir |
| 3 | የማቲዮስ ወንጌል 6:33 ምን ይላል? (Translation: What does Matiwos 6:33 say?) | amh | የማቲዮስ ወንጌል 6:33 ምን ይላል? ወስስስስስስስስስስስስስስስስስስስስስስስ ስስስስስስስስስስስስስስስስስስስስስስ ስስስስስስስስስስ | byn |
| 4 | ሃቢባ አብ ኢትዮጵያ ኮምቦልቻ ከተማ ውሻጢ ተወሊዳ ዓቢያ። ናይ ሃቢባ ስራሕ (Translation: Habiba was born and raised in Ethiopia's Kombolcha city. Habiba works as | tir | ሃቢባ አብ ኢትዮጵያ ኮምቦልቻ ከተማ ውሻጢ ተወሊዳ ዓቢያ። ናይ ሃቢባ ስራሕስስስስስስስስስስስስስስስስስስ ስስስስስስስስስስስስስስስስስስስስ ስስስስስስስስስስስ | byn |

Figure 10: Generation examples for Amharic and Tigrinya using BLOOM model. We see that the generations for Tigrinya and Amharic have repeated characters that make for long strings; such generations are mostly misclassified as Bangla(byn). On the other hand, even in cases where the generated text is predicted to be in the same language as the prompt language, there are artifacts like 'emailuser' that appear in the generated text for these languages.

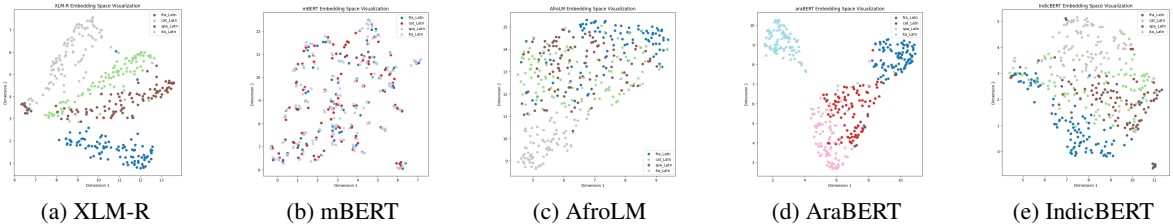

(a) XLM-R      (b) mBERT      (c) AfroLM      (d) AraBERT      (e) IndicBERT

Figure 11: Last layer autoencoder embeddings for Romance languages. We see that XLM-R and AraBERT form some language-specific clusters while the rest have mixed representations for the languages.

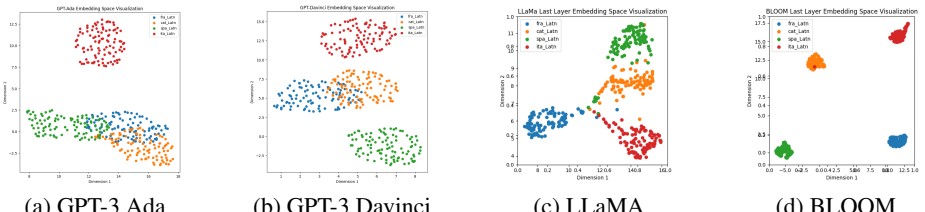

(a) GPT-3 Ada      (b) GPT-3 Davinci      (c) LLaMA      (d) BLOOM

Figure 12: Learned representations for autoregressive models for Romance languages. All models form some language-specific clusters while BLOOM forms the most distinct clusters for all lanaguegs.

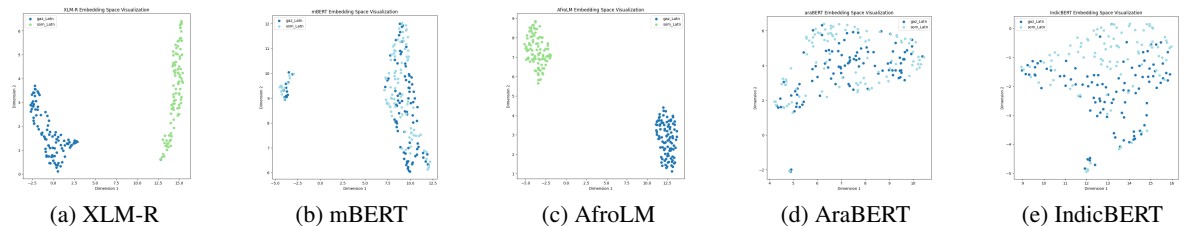

(a) XLM-R      (b) mBERT      (c) AfroLM      (d) AraBERT      (e) IndicBERT

Figure 13: Last layer autoencoder representations for Cushtic languages. We observe clearer separation in XLM-R and AfroLM, while all other models have mixed representations.

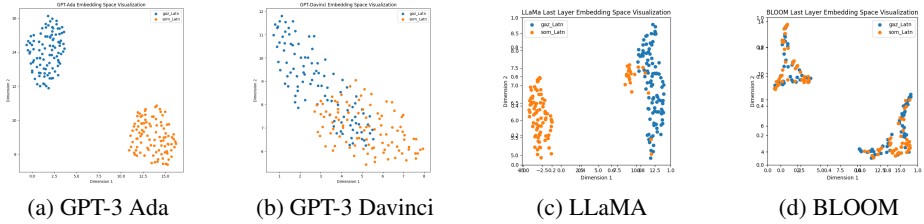

(a) GPT-3 Ada      (b) GPT-3 Davinci      (c) LLaMA      (d) BLOOM

Figure 14: Learned representations of autoregressive models for Cushtic languages. Except for GPT-Ada, all other models have mixed representations.

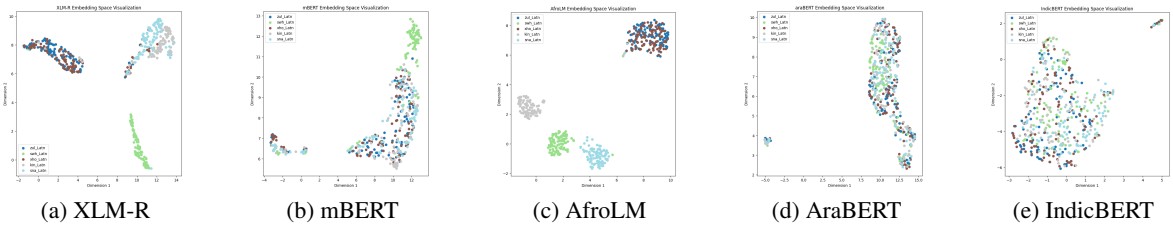

(a) XLM-R      (b) mBERT      (c) AfroLM      (d) AraBERT      (e) IndicBERT

Figure 15: Last layer autoencoder representations for Bantu languages. We observe language-specific clusters for AfroLM with isZulu and Xhosa mixing, while XLM-R forms a clear cluster for Swahili and somewhat mixed clusters for the other languages. The other models mix all the languages.

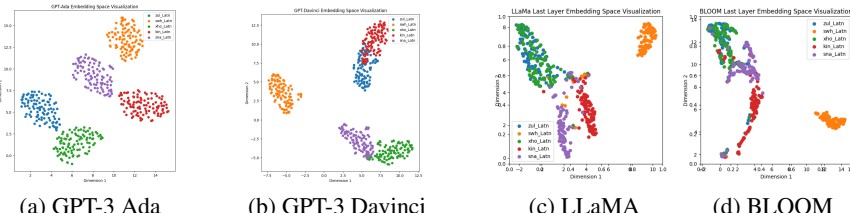

(a) GPT-3 Ada      (b) GPT-3 Davinci      (c) LLaMA      (d) BLOOM

Figure 16: Learned representations from autoregressive models for Bantu languages. We observe that LLaMA and BLOOM have mixed representations for the Bantu languages, while there is a small cluster for some of the Swahili sentences. While GPT-3 Ada separates out all languages, GT3-Davinci has some mix for Shona and Kinyarwanda and mix for isZulu and Xhosa.

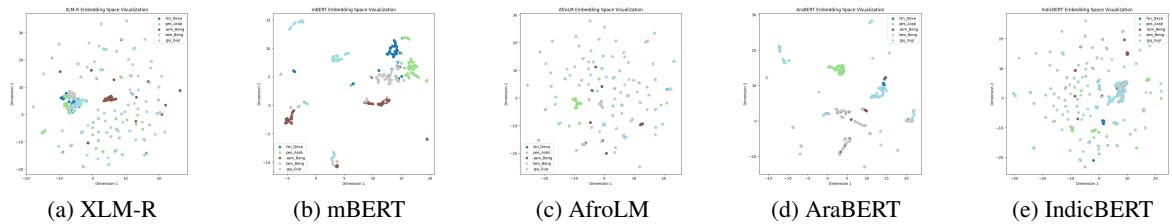

(a) XLM-R      (b) mBERT      (c) AfroLM      (d) AraBERT      (e) IndicBERT

Figure 17: Last layer representations for Indo-Iranian languages. All models have mixed representations for all the Indo-Iranian languages.

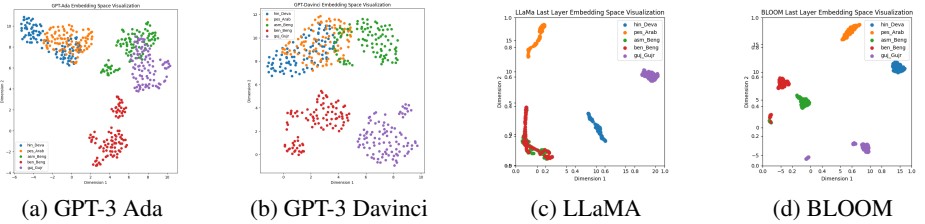

(a) GPT-3 Ada      (b) GPT-3 Davinci      (c) LLaMA      (d) BLOOM

Figure 18: Learned representations from autoregressive models for Indo-Iranian languages. Here, we see LLaMA and BLOOM have some language specific clusters with LLaMA mixing Bengali and Assamese. GPT-3 Davinci mixed Persian, Hindi, and Assamese, while GPT-3 Ada mixed Persian and Hindi and also mixed Gujarati and Assamese.

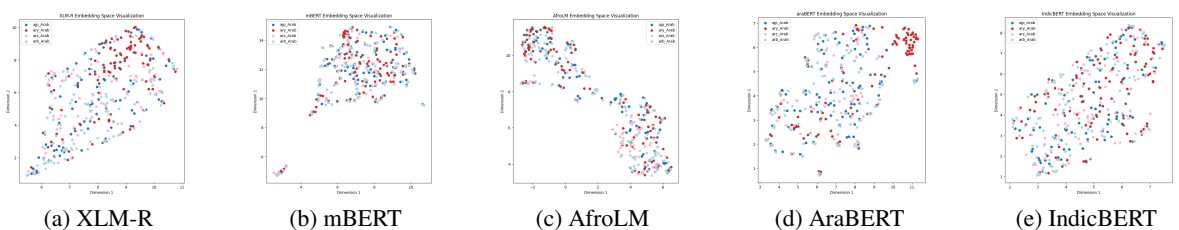

(a) XLM-R      (b) mBERT      (c) AfroLM      (d) AraBERT      (e) IndicBERT

Figure 19: Last layer representations for the Arabic dialects. All models have mixed representations for all dialects, while we observe a small cluster for Egyptian Arabic in AraBERT.

| Languages | Family | No. of speaker |
|---|---|---|
| Arabic (ara) | Afro-Asiatic/ Semitic | 630M |
| Amharic (amh) | Afro-Asiatic/ Semitic | 57M |
| Tigrinya (tir) | Afro-Asiatic/ Semitic | 9M |
| Afaan Oromo (orm) | Afro-Asiatic/ Cushitic | 37M |
| Somali (som) | Afro-Asiatic/ Cushitic | 22M |
| French (fra) | Indo-European/ Romance | 350M |
| Catalan (cat) | Indo-European/ Romance | 9.2M |
| Spanish (spa) | Indo-European/ Romance | 595M |
| Italian (ita) | Indo-European/ Romance | 68M |
| Hindi (hin) | Indo-European/ Indo-Iranian | 592M |
| Iranian Persian (per) | Indo-European/ Indo-Iranian | 81M |
| Assamese (asm) | Indo-European/ Indo-Iranian | 15M |
| Bengali (ben) | Indo-European/ Indo-Iranian | 234M |
| Gujarati (guj) | Indo-European/ Indo-Iranian | 56M |
| isZulu (zul) | Niger-Congo/Bantu | 26M |
| Swahili (swa) | Niger-Congo/Bantu | 88M |
| isiXhosa (xho) | Niger-Congo/Bantu | 19M |
| Kinyarwanda (kin) | Niger-Congo/Bantu | 15M |
| Shona(sna) | Niger-Congo/Bantu | 17.8M |
| Chinese (zho) | Indo-Chinese (Sino-Tibetan) | 1.35 B |

Table 6: Languages included in the experiment.

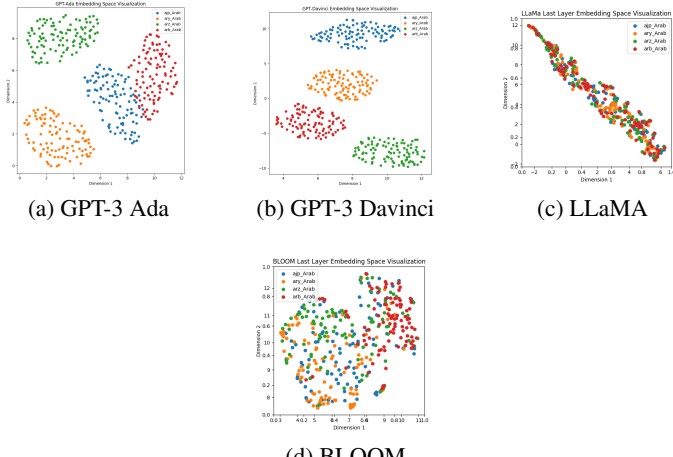

(a) GPT-3 Ada     (b) GPT-3 Davinci     (c) LLaMA

(d) BLOOM

Figure 20: Learned representations from autoregressive models for Arabic Dialects. Both BLOOM and LLaMA models have mixed representations for all dialects, while GPT models form dialect-specific clusters.

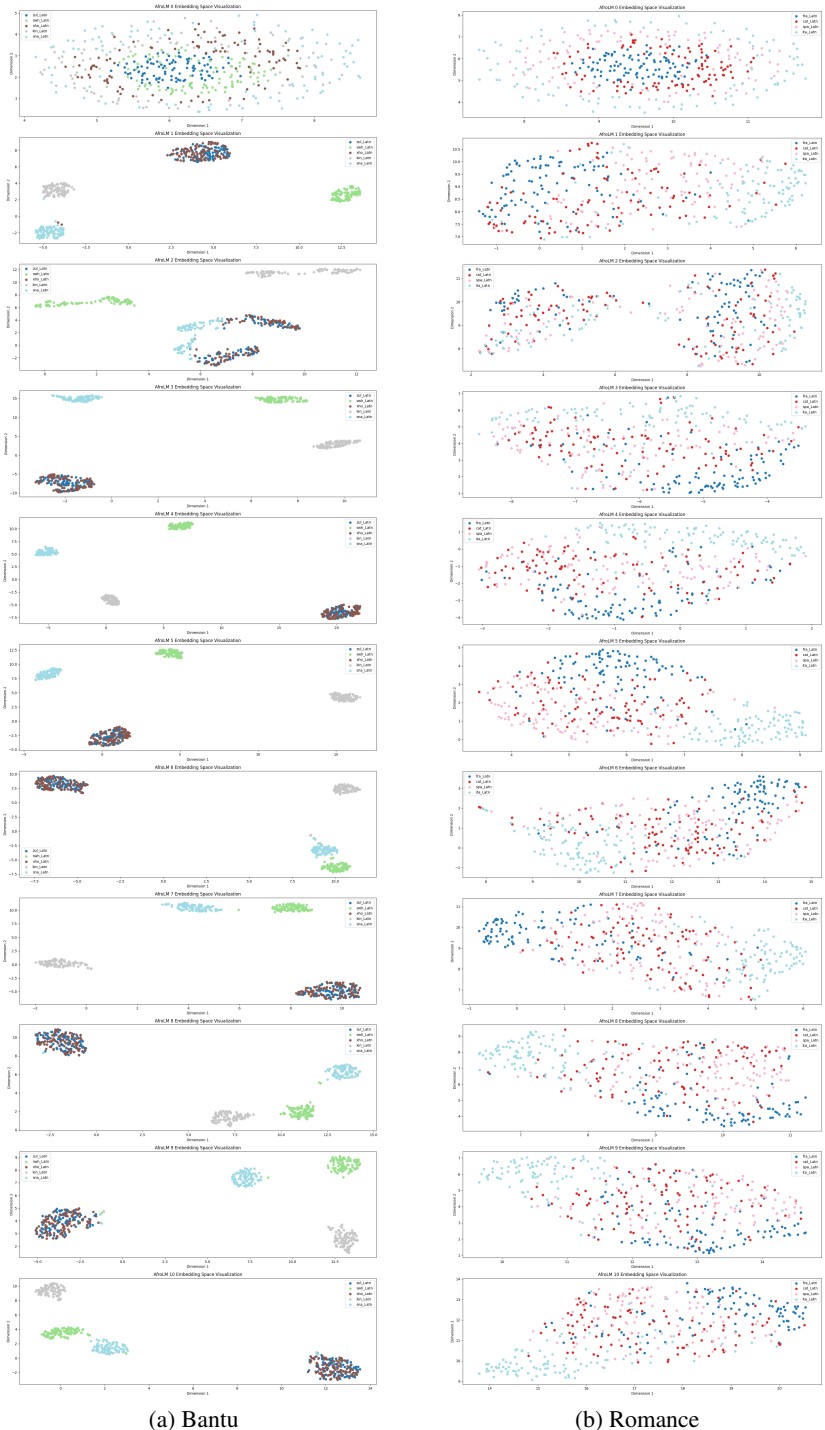

(a) Bantu          (b) Romance

Figure 21: AfroLM representations on all layers for Bantu and Romance Languages. For Bantu languages, we observe that Xhosa and isZulu occupy the same cluster while all other languages form independent clusters. We observe that in the middle layers, the individual clusters are further away from each other and start spreading closer in the last layers. For Romance languages, we observe that all languages are mixed with no clear cluster formed for any of the languages.

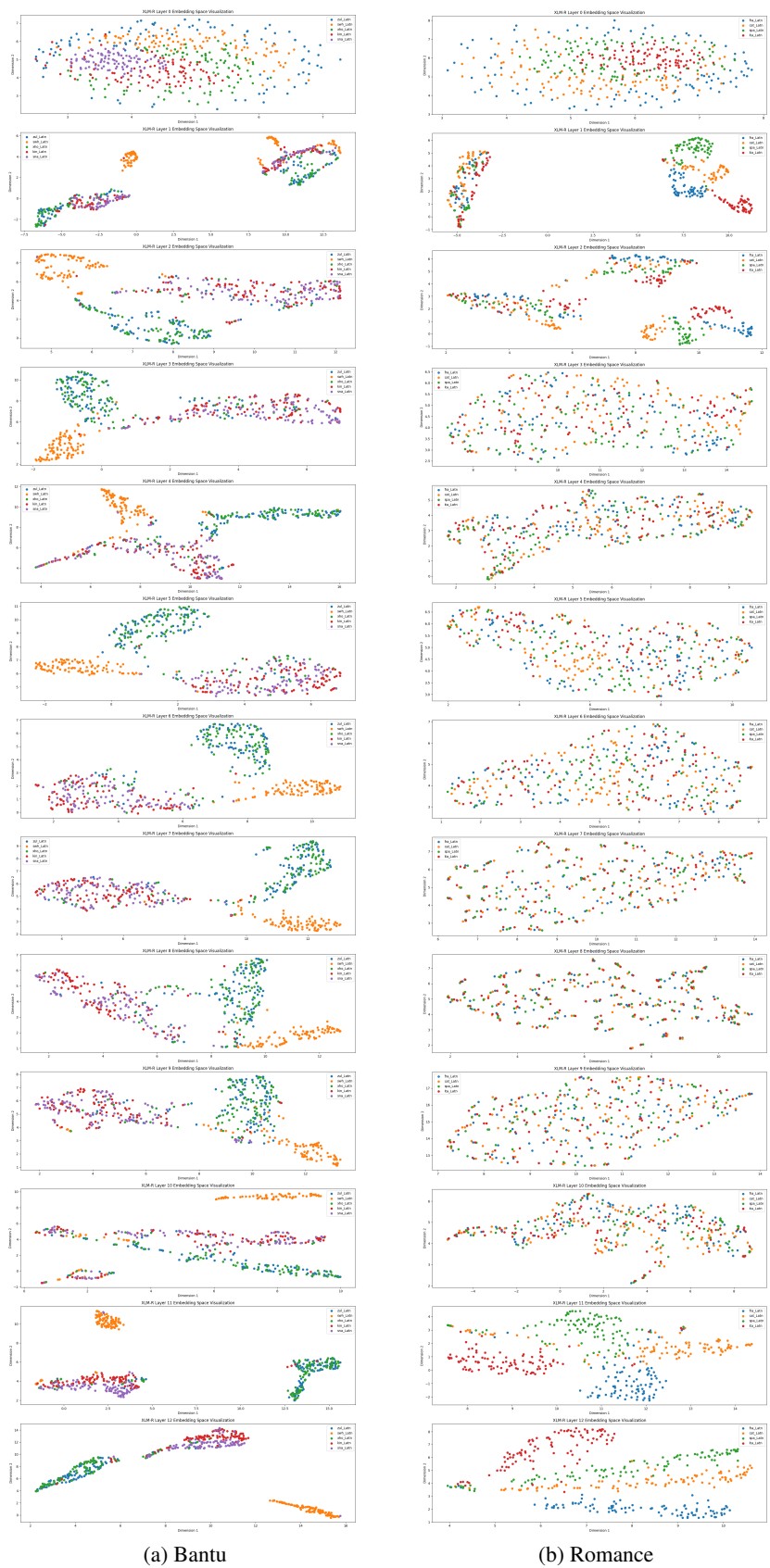

(a) Bantu        (b) Romance

Figure 22: XLM-R representations on all layers for Bantu and Romance Languages. We observe for Romance languages, the representations in the middle layers, particularly layers 8, 9, and 10, exhibit semantic clusters, and the last layers show language-specific clusters. For Bantu languages, we observe some language-specific clusters in the middle layers, with Kinrwanda and Shona representations mixing as well as Xhosa and isXulu forming a pair. Only Swahili forms an independent cluster for Bantu languages.