# OpenReview forum: "The Less the Merrier? Investigating Language Representation in Multilingual Models"
_EMNLP/2023/Conference — EMNLP 2023 Findings_

### Official Review · Reviewer_U2Y9 · 2023-08-01

**Soundness:** 3

**Excitement:**

2: Mediocre: This paper makes marginal contributions (vs non-contemporaneous work), so I would rather not see it in the conference.

**Missing References:**

Some older and newer references in the area that I did not see in the bibliography. No need to include all of them, but I think that some of the context of what is happening in the area of multilingual model analysis is slightly glossed over in the paper, which leads to the discussion around findings being a bit lacking:

Choenni and Shutova, Cross-neutralising: Probing for joint encoding of linguistic information in
multilingual models

Nekoto et al, Participatory Research for Low-resourced Machine Translation: A Case Study in African Languages

 Ògúnrèmí et al, Mini But Mighty: Efficient Multilingual Pretraining with Linguistically-Informed Data Selection

Lauscher et al From Zero to Hero: On the Limitations of Zero-Shot Language Transfer with Multilingual Transformers

Papadimitriou et al Deep Subjecthood: Higher-Order Grammatical Features in Multilingual BERT

Artetxe et al On the Cross-lingual Transferability of Monolingual Representations

Conneau et al Emerging Cross-lingual Structure in Pretrained Language Models

Wang et al On Negative Interference in Multilingual Models: Findings and A Meta-Learning Treatment

Libovicky et al, . How language-neutral is multilingual BERT?

Pires et al How Multilingual is Multilingual BERT?

**Paper Topic And Main Contributions:**

This paper analyzes multilingual language models, examining their training in, their representation of, and their generation of a diverse set of languages. The paper looks at large-scale models largely focused on performance in high-resource languages (like GPT-3), large multilingual models like XLM-R, as well as community-based models focusing on more geographically-focused sets of languages like AfroLM. The findings include a literature analysis of which languages are represented in the language models they choose, as well as three sets of experiments: 1) looking at how sentences in different languages cluster in embedding space visualizations 2) classifying languages from the embedding space of different multilingual models to better understand representation and 3) Seeing how multilingual models can generate in different languages given different prompts that prompt towards a language.

**Reasons To Accept:**

This paper deals with an important and timely question, and provides an interesting overview to the issue of language representation and separation in multilingual models.

The fact that the authors consider subtleties like dialect, script, and language similarity is very cool, and I really like how this has the potential to deepen the analyses around multilingual models, and the many parts of section 6.1 in this paper are definitely a nice start towards this deepening.

I think that the results in Section 6.2/Table 3 are very interesting, and I like the authors’ analysis about community-based models and what the results are in terms of low-resource language acuity.

**Reasons To Reject:**

Though the motivation of the paper is very interesting, and the clustering language separation experiments are intriguing, I think that many parts of the paper do not have very robust experimental methodologies or exciting findings.

I think that the visualization experiments are possibly a bit shaky. Though t-SNE is a useful heuristic  for understanding distance in high-dimensional space, I don’t think that basing empirical results off of analyzing t-SNE plots is that robust. For example, there might be rich language separation spaces in the original space that t-SNE collapses down because semantic relationships seem more important in maintaining distance. Furthermore, it would have been nice to see more analysis or discussion about what embedding-based results can tell us about multilinguality in general. It’s not clear what kind of embedding spaces translate to good multilingual models: for example, if models had the embeddings of different languages in very separated smalls subspaces, would this be better than if models reserved one dimension for language identification and used d-1 dimensions for the parallel semantic space? I think that the meaning of embedding-based results is a bit unclear without further analysis, and it would have been nice to have read some more discussion on this.

This next point might be more a matter of taste, but I’m not sure that Section 4 is necessary. It’s definitely interesting to map out language models in this way, but a lot of space is taken up treating this investigation as a set of findings, when it’s simply an aggregation of numbers pretty clearly reported by the model creators. Audits of models can of course be very interesting, especially given how many new models are coming out. But I do not feel that the investigation here had any framing or results that gave me new ways of thinking about the topic, or that were very in-depth and systematic between models in a way that will be useful for reference.

Overall, the question of the acuity of multilingual representation in multilingual models has a rich literature, that is slightly undercited in this paper (see citations), and I’m not sure that this paper fills in a very robust answer for any of the questions posed in this line of literature. That being said, some of the findings and plots are interesting, and could provide good data points for future research.

**Reproducibility:**

3: Could reproduce the results with some difficulty. The settings of parameters are underspecified or subjectively determined; the training/evaluation data are not widely available.

**Reviewer Confidence:**

3: Pretty sure, but there's a chance I missed something. Although I have a good feel for this area in general, I did not carefully check the paper's details, e.g., the math, experimental design, or novelty.

**Typos Grammar Style And Presentation Improvements:**

Suggestions:

Figure 1 is a bit unclear to me, and I think that there are more striking ways to present this data. From what I understand, the existence of an orange indicates whether the language is in XLM-R. However, this is a bit asymmetric, as makes it unclear which languages are left out, because “left out” is just signified with a branch without an orange (often obscured by another orange). I think there are better ways to present this data, eg with a simpler more latex-like tree where languages included are one color and languages not included are another. The diagram also does not represent the fact that there are very few non-indoeuropean languages in XLM-R.

line 134 It might be nice to expand on the active learning side of AfroLM for one more sentence.

Figure 2a: I’m not sure the numbers are totally accurate, I checked Nigeria (which has 1 language represented in AfroLM according to the heatmap) and AfroLM has at least Hausa, Igbo, and Yoruba. Maybe I’m misunderstanding the graph, and each language is only counted for one country even if it goes across borders?

I think that section 3.3 (Evaluation Dataset) and section 5 (methods) could be together in one  methodology section. Currently, they’re separated by a section when they’re very related.

It would be great for some appendix graphs (especially those mentioned in lines 279, 283, 285

Fig 3 and Fig 5 could maybe be ordered the other way around, since the whole paper seems to mention autoencoder models before autoregressive models.

Figs 3, 5, 6: the writing font is pretty small, could consider making it bigger

Small typos:

line 38 Anglo-centeric → Anglo-centric (Or probably better without the hyphen: Anglocentric)

line 283 AforLM → AfroLM

---

> ### Author Rebuttal · Authors · 2023-08-29
>
> We first want to thank the reviewer for their valuable feedback. Regardless of the outcome for this conference, it helped us critique our own work and think of experiments to run to strengthen our arguments. Below, we address issues raised:
>
> 1. Though t-SNE is a useful heuristic for understanding distance in high-dimensional space, I don’t think that basing empirical results off of analyzing t-SNE plots is that robust. For example, there might be rich language separation spaces in the original space that t-SNE collapses down because semantic relationships seem more important in maintaining distance
>
> We first recognize the limitation of using only t-SNE in our analysis. We initially started the experiments as visualizations of the embedding space in multilingual models and had the downstream tasks to see if the visualization observation was matched in the downstream task performance. We then experimented with UMAP as a dimensionality reduction method for some of the language families in our experiments with autoencoder models and did not observe significant divergences in the visualizations. Hence, we did not do further investigation with UMAP. Following reviewers' inquiries, we ran additional experiments for Bantu and Romance language families using UMAP as a reduction technique with all models and observed significant divergence for GPT-3. We saw that for autoencoder models like AfroLM, UMAP showed separate clusters for Kinyarwanda, Swahili and Shona while Zulu and Xosha were mixed; identical to the observation in t-SNE visualizations. However, with GPT-3 embedding, UMAP showed separate clusters for all languages while in t-SNE there were no language independent clusters. We observed similar difference for Romance and Semitic languages when using UMAP on GPT-3 embedding. We will include these results in the updated version of the paper.
>
> 2. Furthermore, it would have been nice to see more analysis or discussion about what embedding-based results can tell us about multilinguality in general. It’s not clear what kind of embedding spaces translate to good multilingual models
>
> We agree with the reviewer that a deeper discussion around the implication of the embedding space in overall multilinguality would be interesting. For instance, looking at the geometrical structure of the embedding space and figuring out different dimensions would allow us to further argue about multilinguality and its relationship to embedding spaces. It is very well possible that, for example in a generative model, the model does generate text consistently in one language with out necessarily having separate embedding clusters for different languages (if current token is in language L1 and the language model has learned good representations regardless of weather separate clusters exist, the next predicted token could also be in language L1) However, we initially did not want to speculate too much on the implication of our results, in fear of making over arching claims about what the embedding clusters tell us. We tried to strengthen our arguments by using the down stream task of text generation and observed for example that the generations for Tigrinya and Amahric are mixed, while the embedding visualizations (both in t-SNE original results and UMAP results we collected after receiving reviews) both show the embeddings for those languages are also mixed. But, correlation does not equate to causation and so we refrained from sharing such speculations without further investigation. We hope to improve up on our work by designing further probing tasks and are willing to include experiments inspired by previous work shared by the reviewer. If accepted, we will add a section in our discussion on speculations and possible implications while also cautioning on the limitations of our work. We do however believe that our results give good jumping of points for future work and that they provide insights to the community in *how* learned representations for different language groups (dialect, language family, writing script) differ across several multilingual models, which will aid future work in understanding *why (and if)* those representations affect downstream performance.
>
> 3. This next point might be more a matter of taste, but I’m not sure that Section 4 is necessary. It’s definitely interesting to map out language models in this way, but a lot of space is taken up treating this investigation as a set of findings, when it’s simply an aggregation of numbers pretty clearly reported by the model creators. Audits of models can of course be very interesting, especially given how many new models are coming out. But I do not feel that the investigation here had any framing or results that gave me new ways of thinking about the topic, or that were very in-depth and systematic between models in a way that will be useful for reference.
>
> We agree with the reviewer that audits of models are important for NLP research. We argue that the inclusion of section 4 is crucial to our work in that it shows which languages are included and which languages are excluded from mainstream NLP models. We argue that it prevents a monolithic understanding of low-resourced-ness and avoids ‘sample/token inclusion’ of certain languages in studies as representative of all low-resourced languages. Further, it critiques language models that are built with a focus on low-resourced languages and shows how there is still a long way to go (see Fig. 2, p.4). We also argue that our analysis is inline with the theme track topic: Large Language Models and the Future of NLP.
>
> 4. Overall, the question of the acuity of multilingual representation in multilingual models has a rich literature, that is slightly under-cited in this paper (see citations), and I’m not sure that this paper fills in a very robust answer for any of the questions posed in this line of literature.
>
> We thank the reviewer for their pointers to works in the multilingual representation space and will include the citations in our camera-ready paper. We argue that our work provides insights for the question of multilingual representation in low-resourced language settings and provides jumping-off points for future research in this area. Further, with the integration of comments from all of the reviewers, we argue our work becomes stronger and provides further evidence to support our claims. We welcome any feedback the reviewer might have to make our paper even stronger.
>
> 5. That being said, some of the findings and plots are interesting, and could provide good data points for future research
>
> We thank the reviewer for this positive feedback and will continue to improve upon this work regardless of the outcome:-)
>
> We are also grateful for the pointers on the typos, presentation and grammar issues. We will correct those in the updated paper.

---

### Official Review · Reviewer_NTN2 · 2023-08-04

**Typos Grammar Style And Presentation Improvements:** 283
**Soundness:** 3

**Excitement:**

3: Ambivalent: It has merits (e.g., it reports state-of-the-art results, the idea is nice), but there are key weaknesses (e.g., it describes incremental work), and it can significantly benefit from another round of revision. However, I won't object to accepting it if my co-reviewers champion it.

**Missing References:**

Scao, T. L., Fan, A., Akiki, C., Pavlick, E., Ilić, S., Hesslow, D., ... & Manica, M. (2022). Bloom: A 176b-parameter open-access multilingual language model. arXiv preprint arXiv:2211.05100.

**Paper Topic And Main Contributions:**

In this paper, they analyze the linguistic coverage and representation of multilingual Language Models (LMs), including autoencoder and auto-regressive models. Firstly they research on the language coverage of different multilingual models, comparing models from big tech companies and community-driven models for a single language family or from a certain continent/region. Secondly, they analyze the representation of those language families across the models. They complement the analysis by testing the performance on language identification and text generation tasks. They also conclude that community-centered models identify languages and dialects better, and much better for low-resource languages.

The paper's main contribution is that it contributes towards our understanding and awareness of the scope of linguistic coverage of current multilingual LLMs.

**Questions For The Authors:**

Question A: Do you have results on your generation tasks for the BLOOM model? Is there any reason you left BLOOM out of the analysis? I think it would add a lot to the analysis to see if we can say "community-centered models are better" (or not) also in autoregressive models.

Note A (No need to reply to this): In case you haven't, and you are planning to evaluate a community-centered LLM, you could also use "GPT-SW3: An Autoregressive Language Model for the Nordic Languages" (which could not be included in your paper, because it was published very recently).

Question B: Have you evaluated these models (at least the autoencoder ones), on an NLU downstream task like NER or similar? (I understand if you have not, as most of us have limited resources of GPUs and people/time, thus, leaving it out of the scope of this work).

**Reasons To Accept:**

- It contributes towards our understanding of the scope of linguistic coverage of current multilingual LLMs.

- It raises awareness about the poor health of many low-resource languages that are left out of the current LLM wave.

- It provides a unified analysis of the representation of languages inside different LMs.

- Taking into account the linguistic families, dialects and scripts to interpret the results is very insightful.

- It proposes a simple evaluation method and shares prompts easily and cheaply extendable to other low-resource languages and models.

- The paper is easy to follow and well organized.

**Reasons To Reject:**

- Among the autoregressive language models, I missed the comparison between LLMs like llama/GPT3 created by big tech companies with the community-driven model BLOOM (Scao et al., 2022) as the authors do among the autoencoder models. The paper says (at 433-434) that "For autoregressive models, we did not have community-centred models to compare with", and I think that BLOOM falls in that community-centred model category (even if it does not include all the languages selected for evaluation). But I think the paper would greatly benefit from comparing BLOOM with the other LLMs on their evaluation, for some of the low-resource languages from bloom (which include many Indic and African languages among the selected 46 languages). If the model is not included in the comparison for completeness, the authors should at least reference it in related work.

Scao, T. L., Fan, A., Akiki, C., Pavlick, E., Ilić, S., Hesslow, D., ... & Manica, M. (2022). Bloom: A 176b-parameter open-access multilingual language model. arXiv preprint arXiv:2211.05100.

- The tasks selected to evaluate the performance of language models are interesting, as they do not require big annotated datasets, but they only require a shallow understanding of a language (I'm not an expert at linguistics, so I might be wrong). For example, in language identification, the presence of unique stopwords of that language, certain script or language-related accents or characters (like Ø for Scandinavian languages) can be enough to identify the languages. Including other tasks which require deeper language understanding capabilities for autoencoder models like for example NER, would help to discern between shallow coverage of low-resource languages, with a deeper understanding of the language. Datasets such as masakhaNER (Adelani et al., 2021) or WikiANN (Pan et al., 2017) could be employed for that evaluation.

Adelani, D. I., Abbott, J., Neubig, G., D'souza, D., Kreutzer, J., Lignos, C., ... & Osei, S. (2021). MasakhaNER: Named Entity Recognition for African Languages. Transactions of the Association for Computational Linguistics, 9, 1116-1131.

Pan, X., Zhang, B., May, J., Nothman, J., Knight, K., & Ji, H. (2017, July). Cross-lingual name tagging and linking for 282 languages. In Proceedings of the 55th Annual Meeting of the Association for Computational Linguistics (Volume 1: Long Papers) (pp. 1946-1958).

**Reproducibility:**

4: Could mostly reproduce the results, but there may be some variation because of sample variance or minor variations in their interpretation of the protocol or method.

**Reviewer Confidence:**

3: Pretty sure, but there's a chance I missed something. Although I have a good feel for this area in general, I did not carefully check the paper's details, e.g., the math, experimental design, or novelty.

---

> ### Author Rebuttal · Authors · 2023-08-29
>
> We first want to thank the reviewer for their valuable feedback. Regardless of the outcome for this conference, it helped us critique our own work and think of experiments to run to strengthen our arguments. Bellow, we present our response to the reviews:
>
>
> 1. Do you have results on your generation tasks for the BLOOM model? Is there any reason you left BLOOM out of the analysis? I think it would add a lot to the analysis to see if we can say "community-centered models are better" (or not) also in autoregressive models.
>
>
> We agree BLOOM is a great contribution to multilingual NLP and is rooted in collaborative community-oriented research, democratizing the development and deployment of LLMs. However, we did not include it in our analysis as it did not fit our definition of community-centered model as detailed in our abstract (‘’models that focus on languages of a given family or geographical location and are built by communities who speak them’’). After reviewer comments, we ran some experiments with BLOOM to see how it compares to the autoregressive models in our experiments. We ran visualization (using UMAP and t-SNE) for Bantu languages and found that, similarly to LLaMa, BLOOM embeddings show separate clusters for Swahili, while all other Bantu languages are clustered together. But, even in the mixed clusters, BLOOM’s representations showed a little more distinction for individual languages compared to LLaMa. Additionally, we ran the text generation task on bloom in the five languages and found that  BLOOM generated text in the language of the prompt at a higher rate than GPT-3 for French, English and Amahric. We have added the results in the table below. We agree with the reviewer that, since BLOOM explicitly includes low-resourced languages, it is a good addition to our comparisons. Hence, we will run further experiments for all other language groups with BLOOM and include the results in our updated paper.
>
> Model | Language | Correct Language Generated (\%) |
> |:---: |:---: |:---: |
> `GPT-3` |Spanish|  95.83 |
> `GPT-3` |French | 91.67 |
> `GPT-3` | Italian | 91.67 |
> `GPT-3` | English |  90.83 |
> `GPT-3` | Amharic | 76.66|
> `GPT-3` | Tigrinya | 45.00 |
> |:---: |:---: |:---: |
> `BLOOM` | Spanish | 93.33 |
> `BLOOM` | French | 100 |
> `BLOOM` | Italian | 53.33 |
> `BLOOM` | English |100 |
> `BLOOM` | Amharic | 80.00 |
> `BLOOM` | Tigrinya | 33.33 |
>
> 2. Note A (No need to reply to this): In case you haven't, and you are planning to evaluate a community-centred LLM, you could also use "GPT-SW3: An Autoregressive Language Model for the Nordic Languages" (which could not be included in your paper, because it was published very recently).
>
> We thank the reviewer for the pointer to GPT-SW3: An Autoregressive Language Model for the Nordic Languages; we were not aware of this recent model at the time we submitted our paper.
>
> 3. Have you evaluated these models (at least the autoencoder ones), on an NLU downstream task like NER or similar? (I understand if you have not, as most of us have limited resources of GPUs and people/time, thus, leaving it out of the scope of this work)
>
> We agree with the reviewer that evaluating downstream tasks like NER would provide further insight into the performance of different languages and on NLU tasks. However, we scoped our work to focus on understanding and visualizing the embedding of multilingual models and how those distinguish between different language groups. Evaluating on NLU tasks would require fine-tuning the model on further data for the languages of interest; we were interested in the raw representations of the sequences by the models.  However, we did collect results for NER for Bantu languages from the AfroLM paper for AfroLM, XLM-R and mBERT and we fine-tuned araBERT and IndicBERT for the 5 Bantu languages. As detailed in the table below, we found that AfroLM outperformed all models in all languages except for XLM-R in Kinyarwanda, confirming how community-centered models perform better for low-resourced languages even in NLU tasks. Hence, we will add results from experiments for NER and incorporate the reviewer’s valuable comments.
>
> Language (bantu) |XLM-R   | mBERT | AfroLM| araBERT| IndicBERT |
> |:---: |:---: |:---: |:---: | :---: | :---: |
> `zul` | 84.6  | 81.7  |**86.3**|76.9 |67.2|
> `xho` | 87 | 85 |**87.5**|75.8 |75.3 |
> `sna` | 93.6  | 92.4 |**94.4**|73|83.4  |
> `swa` | 87.5  |86.3   |**87.6**|78.9|79.9 |
> `kin` | **73.9** | 70.9   |72.8|69.3|71.1|

---

### Official Review · Reviewer_6SvF · 2023-08-04

**Soundness:** 2

**Excitement:**

3: Ambivalent: It has merits (e.g., it reports state-of-the-art results, the idea is nice), but there are key weaknesses (e.g., it describes incremental work), and it can significantly benefit from another round of revision. However, I won't object to accepting it if my co-reviewers champion it.

**Missing References:**

- AfroXLMR - https://arxiv.org/pdf/2204.06487.pdf
- AfriBERTa - https://aclanthology.org/2021.mrl-1.11/
- KinyaBERT - https://aclanthology.org/2022.acl-long.367.pdf
- AfriTeVa - https://aclanthology.org/2022.deeplo-1.14.pdf

**Paper Topic And Main Contributions:**

This paper investigates the representations of languages in community-focused multilingual language models and more "general" language models. They analyze representations based on language families, script and dialects. They observe that:
- Community-focused models perform better at distinguishing between languages for low-resource languages
-  Models that showed separate clusters in embedding space across language families and writing scripts showed the highest F1 score for KNN text classification of languages, compared to other models
- Models struggle with dialect separation

**Questions For The Authors:**

- Why do you use the CLS token for encoder models and the last token for LAMA? Did you experiment with averaging all tokens and other possible options?
- Why did you exclude mT5 and AfriTeVa? You stated there are no autoregressive community-focused models, but AfriTeVa exists.

**Reasons To Accept:**

- Investigates multilingual representations in both community-focused and more generic models
-

**Reasons To Reject:**

- Only consider T-SNE (which has several limitations) as a dimensionality reduction technique
- Did not give reason for the choice of sequence representations, e.g. why last token for LAMA?
- Missing important benchmarks like mT5 for an "generic" autoregressive model and AfriTeVa for a community-focused autoregressive models
- Insufficient contributions for a long paper.
- Unclear methodology for the text classification and text generation experiments.

**Reproducibility:**

4: Could mostly reproduce the results, but there may be some variation because of sample variance or minor variations in their interpretation of the protocol or method.

**Reviewer Confidence:**

5: Positive that my evaluation is correct. I read the paper very carefully and I am very familiar with related work.

---

> ### Author Rebuttal · Authors · 2023-08-29
>
> We first want to thank the reviewer for their valuable feedback. Regardless of the outcome for this conference, it helped us critique our own work and think of experiments to run to strengthen our arguments. Below, we address issues raised in the feedback:
>
> 1.  Why do you use the CLS token for encoder models and the last token for LAMA? Did you experiment with averaging all tokens and other possible options? | Did not give reason for the choice of sequence representations, e.g. why last token for LAMA?
>
> In terms of choice for which tokens to use for sequence representation, we chose the last token for autoregressive models following previous work (Neelakantan et. al 2022). For autoencoder models, we used CLS token hidden states as that is used as a sentence representation in BERT based models (Devlin 2019). Following the valuable inquiries from reviewers we performed additional experiments with strategies like averaging different hidden states of all tokens and found that it reduced with-in cluster distances (distance between individual points in already formed clusters) but retained the overall cluster formations. We will add this observation in our updated paper.
>
> 2. Why did you exclude mT5 and AfriTeVa? You stated there are no autoregressive community-focused models, but AfriTeVa exists. | Missing important benchmarks like mT5 for an "generic" autoregressive model and AfriTeVa for a community-focused autoregressive models
>
> In this work, we did not look at Seq-to-Seq models due to time and resource constraints. Hence, we did not look at AfriTeva or mT5. When we claimed there are no autoregressive models that fit our definition of community-centered model, we were referring to decoder-only models; hence, AfriTeva or mT5 models do not belong to this category. We will clarify this point further in the updated version of the paper. Additionally, we ran experiments with visualization of embedding from AfriTeva and mT5 for Bantu languages after reviewer comments and found no clear clusters for any of the languages using UMAP and t-SNE. We will explore all language families and put the results in the updated paper.
>
> 3. Only consider T-SNE (which has several limitations) as a dimensionality reduction technique
>
> We first recognize the limitation of using only t-SNE in our analysis. We initially started the experiments as visualizations of the embedding space in multilingual models and had the downstream tasks to see if the visualization observation was matched in the downstream task performance. We then experimented with UMAP as a dimensionality reduction method for some of the language families in our experiments with autoencoder models and did not observe significant divergences in the visualizations. Hence, we did not do further investigation with UMAP.  Following reviewers' inquiries, we ran additional experiments for Bantu and Romance language families using UMAP as a reduction technique with all models and observed significant divergence for GPT-3. We saw that for autoencoder models like AfroLM, UMAP showed separate clusters for Kinyarwanda, Swahili and Shona while Zulu and Xosha were mixed; identical to the observation in t-SNE visualizations. However, with GPT-3 embedding, UMAP showed separate clusters for all languages while in t-SNE there were no language independent clusters. We observed similar difference for Romance and Semitic languages when using UMAP on GPT-3 embedding.  We will include these results in the updated version of the paper.
>
> 4. Unclear methodology for the text classification and text generation experiments.
>
> With regards to our explanation of the methodology, we describe the dataset we used in Sec 3.3 and the methodology in Sec. 5.1 and 5.2. Since we do not have a separate subsection detailing the text classification task, we understand how it might be unclear for readers. We will separate out the different evaluation tasks and describe them in more detail in the updated paper. If there is an additional concern over the clarity of our methodology, we are happy to incorporate that feedback as well.
>
> 5. Insufficient contributions for a long paper.
>
> There are limited works in understanding multilingualism from a low-resourced language perspective. We believe our work adds sufficient insights into this space and offers jumping of points to explore LLMs for multilinguallity and low-resource NLP. We also believe that our work aligns well with the Theme Track topic and adds to the conversation. Additionally, we have run additional experiments based on feedback from the reviewers which strengthens our arguments and increases the robustness of our work. As such, we argue that our work adds significant contributions to this space.

---

### Meta-Review · Area_Chair_YmrP · 2023-09-24

**Recommendation:** 3

**Metareview:**

This paper explores the linguistic coverage and representation of multilingual language models (MLMs), comparing models from large tech companies and community-driven models. The paper analyzes how different languages, language families, scripts, and dialects are clustered and classified in the embedding space of various MLMs, using visualization and KNN text classification. The paper also evaluates the text generation ability of MLMs for different languages and prompts. The paper finds that community-centered models have better coverage and performance for low-resource languages and dialects, while large-scale models are more focused on high-resource languages. The paper contributes to the understanding of the language representation and separation in multilingual models and provides a nice starting work towards which embedding spaces could be leveraged for visualizing multilingual models. Furthermore, the authors ran additional experiments addressing the concerns/questions from the reviewers about the choice of t-SNE, comparison with other community centered models like BLOOM, other NLU based downstream tasks and will be updating the paper draft with the findings.

---

### Decision · Program_Chairs · 2023-10-07

**Decision:**

Accept-Findings

**Comment:**

This paper explores the linguistic coverage and representation of multilingual language models (MLMs), comparing models from large tech companies and community-driven models. The paper analyzes how different languages, language families, scripts, and dialects are clustered and classified in the embedding space of various MLMs, using visualization and KNN text classification. The paper also evaluates the text generation ability of MLMs for different languages and prompts. The paper finds that community-centered models have better coverage and performance for low-resource languages and dialects, while large-scale models are more focused on high-resource languages. The paper contributes to the understanding of the language representation and separation in multilingual models and provides a nice starting work towards which embedding spaces could be leveraged for visualizing multilingual models. Furthermore, the authors ran additional experiments addressing the concerns/questions from the reviewers about the choice of t-SNE, comparison with other community centered models like BLOOM, other NLU based downstream tasks and will be updating the paper draft with the findings.